# Real-time experimental control using network-based parallel processing

**Byounghoon Kim[1], Shobha Channabasappa Kenchappa[1], Adhira Sunkara[2], Ting-Yu Chang[1], Lowell Thompson[1], Raymond Doudlah[1], Ari Rosenberg[1]\***

[1]Department of Neuroscience, School of Medicine and Public Health, University of Wisconsin–Madison, Madison, United States; [2]Department of Surgery, Stanford University School of Medicine, Stanford, United States

**Abstract** Modern neuroscience research often requires the coordination of multiple processes such as stimulus generation, real-time experimental control, as well as behavioral and neural measurements. The technical demands required to simultaneously manage these processes with high temporal fidelity is a barrier that limits the number of labs performing such work. Here we present an open-source, network-based parallel processing framework that lowers this barrier. The Real-Time Experimental Control with Graphical User Interface (REC-GUI) framework offers multiple advantages: (*i*) a modular design that is agnostic to coding language(s) and operating system(s) to maximize experimental flexibility and minimize researcher effort, (*ii*) simple interfacing to connect multiple measurement and recording devices, (*iii*) high temporal fidelity by dividing task demands across CPUs, and (*iv*) real-time control using a fully customizable and intuitive GUI. We present applications for human, non-human primate, and rodent studies which collectively demonstrate that the REC-GUI framework facilitates technically demanding, behavior-contingent neuroscience research.

**Editorial note:** This article has been through an editorial process in which the authors decide how to respond to the issues raised during peer review. The Reviewing Editor's assessment is that all the issues have been addressed (see decision letter).

DOI: https://doi.org/10.7554/eLife.40231.001

**\*For correspondence:**
ari.rosenberg@wisc.edu

**Competing interests:** The authors declare that no competing interests exist.

## Introduction

Many areas of neuroscience research require real-time experimental control contingent on behavioral and neural events, rendering and presentation of complex stimuli, and high-density measurements of neural activity. These processes must operate in parallel, and with high temporal resolution. The number of labs that can perform such research is limited by the technical demands required to set up and maintain an appropriate control system. In particular, a control system must balance the need to precisely coordinate different processes and the flexibility to implement new experimental designs with minimal effort. Systems favoring precision over usability can hinder productivity because there is a large overhead to learning esoteric or low-level coding languages, and extensive coding demands slow the development of new paradigms. In contrast, systems favoring usability over precision can limit the complexity of supportable paradigms and the ability to perform experiments with high computational demands. Here we present the Real-Time Experimental Control with Graphical User Interface (REC-GUI) framework, which overcomes technical challenges limiting previous solutions by using network-based parallel processing to provide both system precision and usability.

The REC-GUI framework segregates tasks into major groups such as: (*i*) experimental control and signal monitoring, (*ii*) stimulus rendering and presentation, and (*iii*) external data acquisition for high-density neural recordings. Each major group is executed on a different CPU, with communications

between CPUs achieved using internet protocols. Additional CPUs can be used to support any number of major groups, as needed. The REC-GUI framework aims to overcome technical challenges that hinder productivity and consume lab resources by providing a solution that reduces the time and effort required to implement experimental paradigms. Towards this goal, we developed the REC-GUI framework using cross-platform, high-level programming environments. For example, the GUI used for experimental control and signal monitoring is coded in Python. The framework is inherently modular, so system components can be modified or substituted to meet research needs (e.g., using MATLAB to present visual stimuli, or Python to track an animal's position). Because the REC-GUI framework achieves precise experimental control with high-level programming environments, it has a low barrier to entry and can be customized without professional programmers or low-level coding languages.

We demonstrate the generality of the REC-GUI framework using three distinct applications. The first investigates the neural basis of three-dimensional (3D) vision in non-human primates. This application shows that REC-GUI can support state-of-the-art stimulus display capabilities during computationally demanding behavior-contingent experiments, and seamlessly integrate with external data acquisition systems to precisely align behavioral and neuronal recordings. The second investigates 3D vision in humans. This application shows that REC-GUI can implement real-time experimental control using inputs from multiple external devices, and that the experimental control CPU can also support data acquisition. The third implements behavioral training and assessment of mice in a passive avoidance task. This application shows that REC-GUI can control experiments without a fixed trial structure by automating the transformation of external device inputs (e.g., position information from a video tracker) into control outputs for other devices (e.g., a shock scrambler). Our tests demonstrate that the REC-GUI framework facilitates technically demanding experiments necessary for relating neural activity to behavior and perception in a broad range of experimental modalities.

We provide sample code and hardware configurations as templates for adaptation and customization. The REC-GUI website (accessible from https://rosenberg.neuro.wisc.edu/) includes a discussion forum, GitHub link to the software and user manual, and links to relevant downloads. The REC-GUI framework can reduce time needed for programming experimental protocols, facilitate time to focus on research questions and design, improve reproducibility by simplifying the replication of paradigms, and enable scientific advances by reducing technical barriers associated with complex neuroscience studies.

## Materials and methods

### Approaches to experimental control

There are multiple ways to implement experimental control. Here we benchmark alternatives and identify an option that satisfies the joint needs of high temporal precision, experimental flexibility, and minimizing coding demands. To illustrate differences between approaches, consider the task of mapping the visual receptive field of a neuron in a behaving monkey. To perform this task, the monkey must hold its gaze on a fixation target presented on a screen. While the monkey maintains fixation, the experimenter must control the movement of a visual stimulus, such as a bar, on the screen. The success of the mapping depends upon the monkey maintaining accurate and precise gaze on the target. If the monkey breaks fixation, the stimulus must disappear until fixation is reacquired. The code required to implement this task includes three major processes that interface hardware and software: (*i*) eye position tracker with monitoring routines, (*ii*) visual display for stimulus presentation with functions to account for fixation status, and (iii) interactive experimenter control for moving the stimulus and changing parameters in real time.

A *serial processing framework* executes these processes serially within a while-loop (*Figure 1A*). As a consequence of serial processing, every iteration of the loop has a pause in eye position monitoring while the stimulus-related and user-related processes finish. This delay can be problematic, especially if the stimulus rendering demands are high. For example, if rendering and presenting the stimulus takes longer than one cycle of the eye position monitoring process, eye position data will be lost, and the accuracy of the gaze-contingent control compromised. Using this problem as an example of the challenges that arise for a control system, we next consider a solution using multithreading.

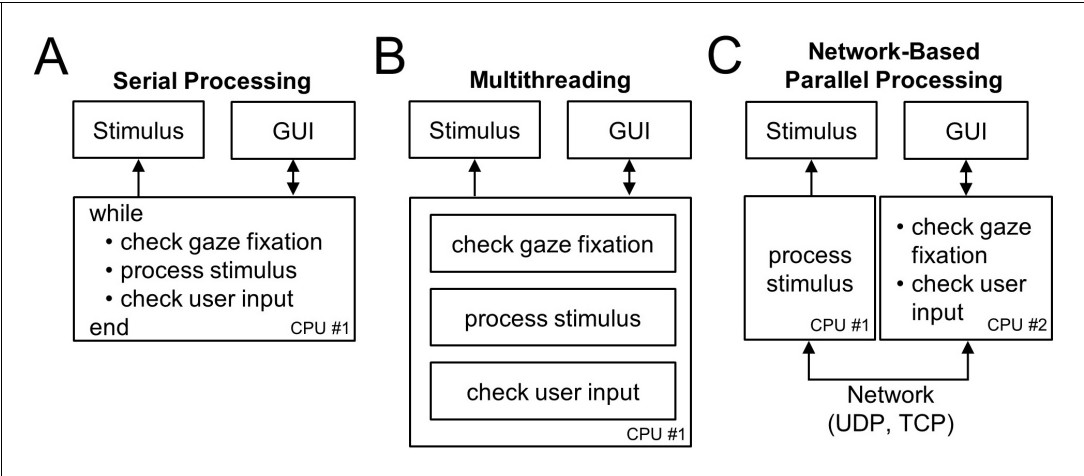

**Figure 1.** Experimental control frameworks. (**A**) Serial processing: All processes are executed serially in a while-loop. (**B**) Multithreading: A single CPU executes multiple processes in parallel on different threads. (**C**) Network-based parallel processing: Multiple processes are executed in parallel on different CPUs coordinated over a network. Individual processes in CPU #2 can be implemented serially or using multithreading. Arrows indicate the direction of information flow.

DOI: https://doi.org/10.7554/eLife.40231.002

A *multithreading framework* provides a solution to this problem by allowing the CPU to execute multiple processes concurrently (*Figure 1B*). Specifically, by separating the eye position monitoring, stimulus-related, and user-related processes onto independent threads, a CPU can execute the processes in parallel such that eye position monitoring can proceed without having to wait for the stimulus- and user-related processes to finish. However, some major coding environments, such as MATLAB, do not currently support multithreading for customized routines. This limitation in multithreading together with the high system demands of such environments can limit real-time experimental control capabilities. Furthermore, since all tasks are implemented on a single CPU, unresolvable system conflicts may arise if different hardware components are only compatible with certain operating systems.

A *network-based parallel processing framework* provides a versatile solution to this problem by dividing experimental tasks across multiple CPUs (*Figure 1C*). This allows tasks to be executed as parallel processes even if multithreading is not supported. Thus, a major benefit of the REC-GUI framework is that challenges arising from limited multithreading support can be resolved without sacrificing the development benefits of software packages that minimize the need for low-level coding knowledge, such as Psychtoolbox which is widely used for stimulus generation (*Brainard, 1997*; *Pelli, 1997*; *Kleiner et al., 2007*). This is particularly valuable if computationally demanding, real-time stimulus rendering is required. A further benefit not possible with multithreading on a single CPU is that different task components can be implemented using different coding languages and on different operating systems. This feature is especially beneficial since research often requires multiple distinct system components. In two applications, we highlight this versatility by using MATLAB to render and present stimuli with Psychtoolbox 3 on one CPU, and Python to run the GUI that implements experimental control on a second CPU. With this configuration, information about changes in fixation status and user inputs to the GUI are relayed via network packets to MATLAB which updates the stimulus accordingly. This ensures that effectively no cycles of eye position data are lost since sending a network packet takes microseconds. More broadly, network-based parallel processing allows the REC-GUI framework to support a wide range of experimental preparations, as long as the system components support network interfacing.

## Overview of the REC-GUI framework

Experimental control is implemented in the REC-GUI framework using network-based parallel processing. The framework divides tasks into major groups, such as: (*i*) experimental control and signal monitoring, (*ii*) stimulus rendering and presentation, and (*iii*) external data acquisition. The number

of groups and how tasks are divided between them is determined based on experimental needs. Different groups are executed on separate CPUs that communicate through internet protocols: user datagram protocol (UDP) and transmission control protocol (TCP). The choice of where to use UDP or TCP depends on the task demands. UDP is fast because it does not perform error checking (i.e., processing continues without waiting for a return signal confirming if a data packet was received). TCP is slower than UDP, but more reliable because it uses error checking and temporally ordered data transmission. To demonstrate the generality of the REC-GUI framework, we implemented three distinct applications using different system configurations.

### Three example applications

*Application 1: Neural basis of 3D vision in non-human primates.* A 3D orientation discrimination task was performed by a rhesus monkey (*Macaca mulatta*). Surgeries and procedures were approved by the Institutional Animal Care and Use Committee at the University of Wisconsin–Madison, and in accordance with NIH guidelines. A male rhesus monkey was surgically implanted with a Delrin ring for head restraint. At the time of the procedure, the monkey was ~4 years of age and 7 kg in weight. After recovery, the monkey was trained to fixate a visual target within 2° version and 1° vergence windows using standard operant conditioning techniques.

The monkey was then trained to perform an eight-alternative 3D orientation discrimination task (*Figure 2A*). In the task, the monkey viewed 3D oriented planar surfaces, and reported the direction of planar tilt with a saccadic eye movement to an appropriate choice target. Planar surfaces were

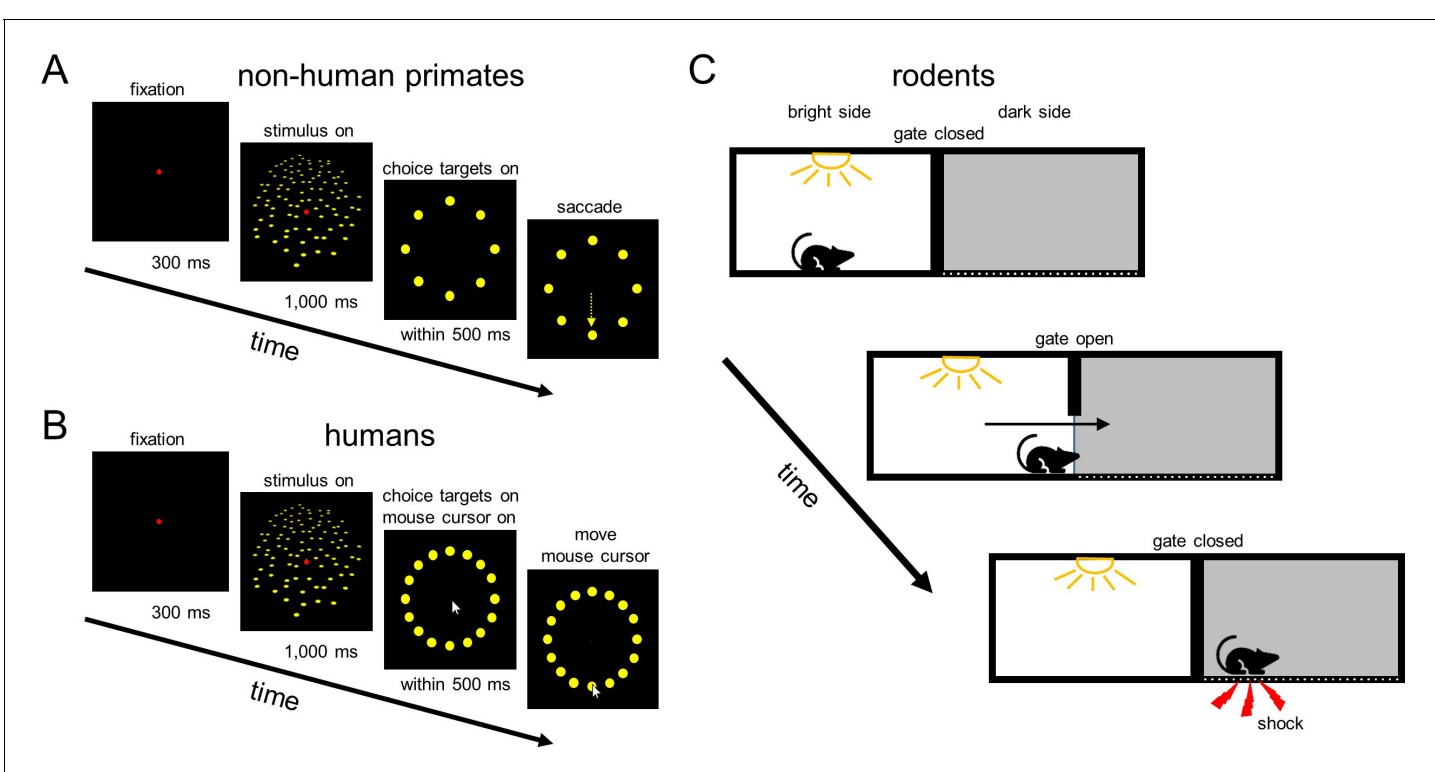

**Figure 2.** Behavioral tasks. (A) Application 1: Neural basis of 3D vision in non-human primates. A monkey fixated a target (red) at the screen center for 300 ms. A planar surface was then presented at one of eight tilts for 1000 ms (one eye's view is shown) while fixation was maintained. The plane and fixation target then disappeared, and eight choice targets corresponding to the possible tilts appeared. A liquid reward was provided for a saccade (yellow arrow) to the target in the direction that the plane was nearest. (B) Application 2: 3D vision in humans. The task was similar to the monkey task, but there were sixteen tilts/targets, and choices were reported using the cursor of a computer mouse that appeared at the end of the stimulus presentation. In *A* and *B*, dot sizes and numbers differ from in the actual experiments. (C) Application 3: Passive avoidance task in mice. A mouse was placed in a bright room that was separated from a dark room by a gate. After the gate was lifted, the mouse entered the dark room. This was automatically detected by the GUI which triggered an electric shock through the floor of the dark room.
DOI: https://doi.org/10.7554/eLife.40231.003

presented at tilts ranging from 0° to 315° in 45° steps, and slants ranging from 15° to 60° in 15° steps. A frontoparallel plane (tilt undefined, slant = 0°) was also presented. The surfaces were defined as random dot stereograms with perspective and stereoscopic cues (N = 250 dots; dots subtended 0.3° of visual angle at the screen distance of 57 cm). At the start of each trial, the monkey fixated a target at the center of an otherwise blank screen for 300 ms. A 20° stimulus centered on the target was then presented for 1,000 ms while fixation was maintained. The stimulus and fixation target then disappeared, and eight choice targets appeared at a radial distance of 11.5° with angular locations of 0° to 315° in 45° increments (corresponding to the possible planar tilts). The monkey was rewarded with a drop of water or juice for choosing the target in the direction that the plane was nearest. If fixation was broken before the appearance of the choice targets, the trial was aborted.

To confirm the ability to accurately and precisely align stimulus-related, behavior-related, and neuronal events using the REC-GUI framework, we measured the 3D orientation tuning of neurons in the caudal intraparietal (CIP) area while the monkey performed the task (*Rosenberg et al., 2013*; *Rosenberg and Angelaki, 2014a*; *Rosenberg and Angelaki, 2014b*). A tungsten microelectrode (~1 MΩ; FHC, Inc.) was targeted to CIP using magnetic resonance imaging scans. The CARET software was used to segment visual areas, and CIP was identified as the lateral occipitoparietal zone (*Van Essen et al., 2001*; *Rosenberg et al., 2013*). A recording grid for guiding electrode penetrations was aligned with the MRI in stereotaxic coordinates using ear bar and grid markers (*Laurens et al., 2016*). Neuronal responses were sampled and digitized at 30 Khz using a Scout Processor (Ripple, Inc.). Single-neuron action potentials were identified by waveform (voltage-time profile) using the Offline Sorter (Plexon, Inc.).

*Application 2: 3D vision in humans.* A variation of the 3D orientation discrimination task was performed by an adult human male with corrected-to-normal vision (*Figure 2B*). The experimental procedures were approved by the University of Wisconsin–Madison Institutional Review Board, and carried out in accordance with the Declaration of Helsinki. Planar surfaces were presented at sixteen tilts (0° to 337.5° in 22.5° steps), with a slant of 60°. The surfaces were 10° and defined as random dot stereograms with perspective and stereoscopic cues (N = 75 dots; dot size = 0.3° at the screen distance of 57 cm). The trial structure matched the monkey experiment. A chin and forehead rest was used to facilitate a stable head position. Fixation was maintained within 3° version and 2° vergence windows. After a stimulus was presented, the fixation target disappeared and a computer mouse cursor appeared at that location. Sixteen choice targets (0° to 337.5° in 22.5° steps) appeared at a radial distance of 6.5°. The participant then used the mouse to select the choice target corresponding to the perceived planar tilt. If fixation was broken before the appearance of the choice targets, the trial was aborted.

*Application 3: Passive avoidance task in mice.* We implemented a passive avoidance task designed to test learning and memory of fear events in rodents (*Coronas-Samano et al., 2016*). The task was performed by a male mouse (C57BL/6, ~5 months of age and 35 g in weight). All procedures were approved by the Institutional Animal Care and Use Committee at the University of Wisconsin–Madison, and in accordance with NIH guidelines. The task was performed in a two-way shuttle box with two rooms (one bright, the other dark) divided by a wall with a gate (*Figure 2C*). A video tracking module provided a top-down view of the bright room. The mouse was placed in the bright room with the gate closed. After 3 min, the gate was manually opened. Mice naturally prefer the dark room, so they are expected to enter it after a short delay. The mouse was removed from the dark room ~ 3 s after entering. This procedure was repeated twice. The shock scrambler was then activated, and the task was repeated three more times. The scrambler was triggered if the mouse left the bright room, delivering a small electric shock (<0.9 mA), and the gate was manually closed. After ~ 3 s of shock delivery, the mouse was removed from the dark room and placed in its home cage for ~5 min. If the mouse did not enter the dark room after 3 min, it was moved to its home cage.

## System configurations for the three applications

*Application 1: Neural basis of 3D vision in non-human primates.* This application implements a 3D discrimination task with neuronal recordings using three major task groups: (*i*) experimental control, signal monitoring, and external device control, (*ii*) stimulus rendering and presentation, and (*iii*) external data acquisition (*Figure 3A*). The application requires the stereoscopic display of stimuli

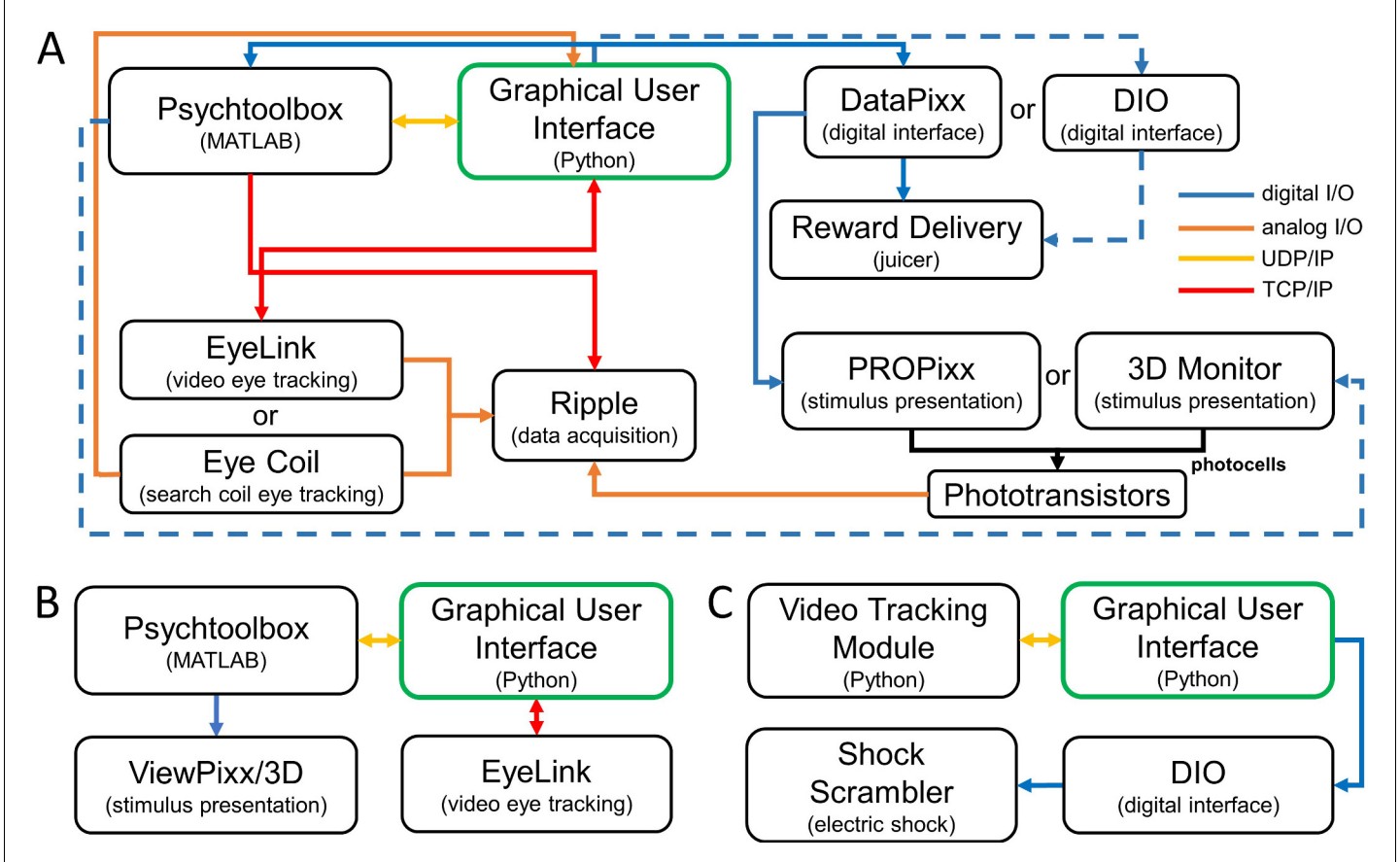

**Figure 3.** Three system configurations using the Real-Time Experimental Control with Graphical User Interface (REC-GUI) framework. Experimental control and monitoring are achieved using a GUI (green box) that coordinates components such as: (i) stimulus rendering and presentation, (ii) external input devices, (iii) external output devices, and (iv) data acquisition. Arrows indicate the direction of information flow. (**A**) Application 1: Neural basis of 3D vision in non-human primates. This application implements a gaze-contingent vision experiment with neuronal recordings. Dashed lines show an alternative communication pathway. (**B**) Application 2: 3D vision in humans. This application implements a gaze-contingent vision experiment without external devices for data acquisition or reward delivery. (**C**) Application 3: Passive avoidance task in mice. This application uses a video tracking module and external shock scrambler to automate behavioral training and assessment.

DOI: https://doi.org/10.7554/eLife.40231.004

The following figure supplements are available for figure 3:

**Figure supplement 1.** Experimental routine and communication flowchart between the stimulus CPU and experimental control CPU showing the exchange of parameters for stimulus rendering and behavioral control for Application 1.

DOI: https://doi.org/10.7554/eLife.40231.005

**Figure supplement 2.** Network configuration for Application 1: Neural basis of 3D vision in non-human primates.

DOI: https://doi.org/10.7554/eLife.40231.006

rendered using multi-view geometry (*Hartley and Zisserman, 2003*), eye tracking for gaze-contingent stimulus presentation and detecting perceptual reports, reward delivery, and neuronal recordings. Visual stimuli were rendered as right and left eye 'half-images' using Psychtoolbox 3 in MATLAB 2017a, and run on Linux (Ubuntu 16.04, Intel Xeon Processor, 24 GB RAM, NVIDIA GeForce GTX 970). A PROPixx 3D projector (VPixx Technologies, Inc.) with circular polarizer was used to rear-project the stimuli onto a screen at 240 Hz, using the 'flip' command (*Kleiner et al., 2007*) to alternately present the appropriate half-image to the right or left eye (120 Hz per eye). To account for the temporal difference between the projector's refresh rate and the execution rate of the MATLAB script, the flip command was set to wait until the next available cycle of the projector refresh. This minimized the variability in the delay between the flip command and the appearance of the stimulus. A second setup using a 3D monitor (LG Electronics, Inc.) and NVIDIA-2 3D Vision Kit

operating at 120 Hz with shutter glasses (run on Windows 10, Intel Xeon processor, 8 GB RAM, NVIDIA Quadro K4000 graphics card) was used to confirm that the framework is robust to system changes (dashed lines in *Figure 3A*). Results from this setup are not presented since they confirm the more stringent testing results from the first setup.

The REC-GUI framework supports eye tracking using video or scleral search coil (*Judge et al., 1980*) methods (*Figure 3A,B*). For this application, we used video tracking with an EyeLink 1000 plus (SR-Research, Inc.). Binocular eye positions were sampled (1 kHz) and digitized by the EyeLink, and the measurements were sent to the GUI through a TCP connection for real-time analysis. The same measurements were converted into an analog signal and transferred to the Scout Processor for offline analysis. For the GUI to perform real-time analysis of eye movements measured with search coils, the analog outputs of the coil system would need to be sampled and digitized using an analog to digital converter (e.g., USB-1608G, Measurement Computing, Inc.). The same outputs would be transferred to the Scout Processor for offline analysis.

A Scout Processor connected to a Windows 10 CPU (Intel Xeon Processor, 8 GB RAM, NVIDIA GeForce GT 720) was used for the acquisition of electrophysiological data and saving analog and digital signals from external devices. The Scout Processor generated a data file with synchronized behavioral and neuronal signals. Other acquisition systems that support network interfacing can be substituted. Since timestamps are generated in the Scout Processor (a standard acquisition system function), no additional synchronization was required to temporally align the input signals. The GUI saved a backup file on the experimental control CPU containing all data that it transmitted and received along with event codes signaling the occurrence of experimental (e.g., fixation point on, stimulus on) and behavioral (e.g., fixation acquired, choice made) events.

Since the application has three major groups, it requires three CPUs: an experimental control CPU, a stimulus CPU, and a data acquisition CPU. A schematic of the experimental routine and communication between the experimental control and stimulus CPUs is shown in *Figure 3—figure supplement 1*. The experimental control CPU uses multithreading to run two threads that: (*i*) monitor and evaluate eye position, and (*ii*) update parameters in real-time using inputs to the GUI. The stimulus CPU uses a while-loop to render stimuli with parameters provided by the GUI, and to perform gaze-contingent stimulus presentation by repeatedly querying the GUI about the fixation status through a UDP connection. Importantly, the UDP connection is established in asynchronous mode so that the while-loop (execution flow) does not pause while waiting for data packets, which could cause dropped frames and time lags. This application thus tests if the framework can accurately and precisely control external hardware under computationally demanding scenarios, and coordinate with external acquisition devices. The major challenge here is the real-time rendering of 3D stimuli while achieving state-of-the-art display capabilities (high frame rate, 240 Hz presentation) without dropped frames, time lags, or compromising the accuracy of the gaze contingencies.

*Application 2: 3D vision in humans.* This application implements a psychophysical task without neural recordings or reward delivery using two major task groups: (*i*) experimental control, signal monitoring, and data acquisition, and (*ii*) stimulus rendering and presentation (*Figure 3B*). A VIEW-Pixx/3D LED display screen (VPixx Technologies, Inc.) operating at 120 Hz with active shutter glasses was used for stereoscopic presentation. As in Application 1, eye tracking was used for gaze-contingent stimulus presentation. However, perceptual reports were made using the cursor of a computer mouse connected to the experimental control CPU. To implement this change, the eye position monitoring routine used in the first application during the perceptual reporting phase of a trial (*Figure 3—figure supplement 1*, 'saccade made' on the bottom left side under Stimulus CPU) was exchanged for a routine that tracks the mouse cursor position and registers 'clicks'. Experimental control in Application 2 thus required two independent external device inputs, one to track eye position and a second to track mouse cursor position. Unlike in Application 1, the experimental control CPU served as the data acquisition system, saving eye position and computer mouse traces in addition to trial information. This application thus tests if the framework can implement real-time experimental control based on inputs from multiple external devices, and support data acquisition without an external system.

*Application 3: Passive avoidance task in mice.* This application implements a passive avoidance task using two major task groups: (*i*) experimental control, signal monitoring, external device control, and data acquisition, and (*ii*) real-time video tracking of mouse position (*Figure 3C*). A shuttle box (ENV-010MC, Med Associates, Inc.) was divided into bright and dark rooms by a gate. The position

of the mouse within the bright room was tracked using a custom real-time video tracking module that we will describe in future work. However, any commercial tracker with analog or digital outputs could be substituted. Briefly, the tracking module segregates an object from the background using color, and estimates position as the center of mass. The module is coded in Python with the OpenCV library (*Bradski and Kaehler, 2008*), and runs on Linux (Raspbian, Raspberry Pi Foundation). A Raspberry Pi B + with Pi camera module v2 was used to track the mouse with 300 x 400 pixel resolution at ~ 42 frames per second. The module sends position information to the GUI, which monitors and records the tracking results as well as the timing of events (e.g., opening of the gate dividing the two rooms, and the mouse entering the dark room). If the GUI detects that the mouse is not in the bright room, it can activate a shock scrambler (ENV-414S, Med Associates, Inc.) to deliver a foot shock through the floor of the dark room. The experimental control CPU was also used for data acquisition. Using the example of behavioral training and assessment, this application thus tests if the framework can automate experimental control in a task without a fixed trial structure by transforming external device inputs into control outputs for other devices.

## System communications

In the REC-GUI framework, system components communicate through four types of connections: UDP, TCP, analog signals, and transistor-to-transistor logic (TTL) pulses. UDP and TCP connections are achieved with network switches. To provide an example of system communications, we use Application 1 since it has the most complicated structure of the three applications tested. Communications between the experimental control CPU, stimulus CPU, EyeLink (eye tracking system), and Scout Processor (data acquisition system) are shown in *Figure 3*, *Figure 3—figure supplement 2*. Two groups of digital event codes are carried over UDP and TCP connections: (*i*) stimulus-related (fixation target on/off, stimulus on/off, choice targets on/off, etc.), and (*ii*) behavior-related (fixation acquired, fixation broken, choice made, etc.). Stimulus-related event codes are triggered in MATLAB and sent to the GUI using UDP and the Scout Processor using TCP. Behavior-related events triggered by the experimental control CPU are sent to MATLAB which relays the codes to the Scout Processor.

Experimental systems often require multiple pieces of specialized hardware produced by different companies. Network-based communications with such hardware must often occur over non-configurable, predefined subgroups of IP addresses. This occurs in Application 1 since both the EyeLink and Scout Processor have non-configurable, predefined subgroups of IP addresses that cannot be routed over the same network interface card (NIC) without additional configuration of the routing tables. In such cases, a simple way to set up communication with the hardware is to use multiple parallel networks with a dedicated network switch for each piece of hardware. For Application 1, this requires two network switches, and both the stimulus and experimental control CPUs require two NICs assigned to different subgroups of IP addresses (*Figure 3—figure supplement 2*). With this configuration, the stimulus and experimental control CPUs can both directly communicate with the EyeLink and Scout Processor.

Digital communications (TTL pulses) are used for control signals. For example, to open a solenoid valve to deliver a liquid reward (Application 1), or trigger a shock scrambler to provide an electric shock (Application 3). TTL pulses can be controlled by MATLAB and generated by a DataPixx (VPixx Technologies, Inc.; solid blue line in *Figure 3A*). They can also be controlled by the GUI and generated by a digital input/output (DIO) interface (e.g., USB-1608G, Measurement Computing, Inc.; dashed blue line in *Figure 3A*, blue line in *Figure 3C*).

## Configurable experimental control GUI

A core component of the REC-GUI framework is an intuitive and fully configurable GUI for experimental control and signal monitoring (*Figure 4*). The GUI contains six main panels: a monitoring window, task control panel, sending panel, receiving panel, data log, and configurable interfaces for specialized tools. To accommodate different experimental paradigms, the GUI provides a simple method for adding and removing parameters from the control set, as well as manipulating the parameters. The GUI was coded in Python 2.7, and run on Linux for all tested applications (Ubuntu 14.04; Applications 1 and 2: Intel i3 Processor, 8 GB RAM, Intel HD500; Application 3: Core 2 Duo 2.80 GHz, 4 GB RAM, NVIDIA GeForce4 MX).

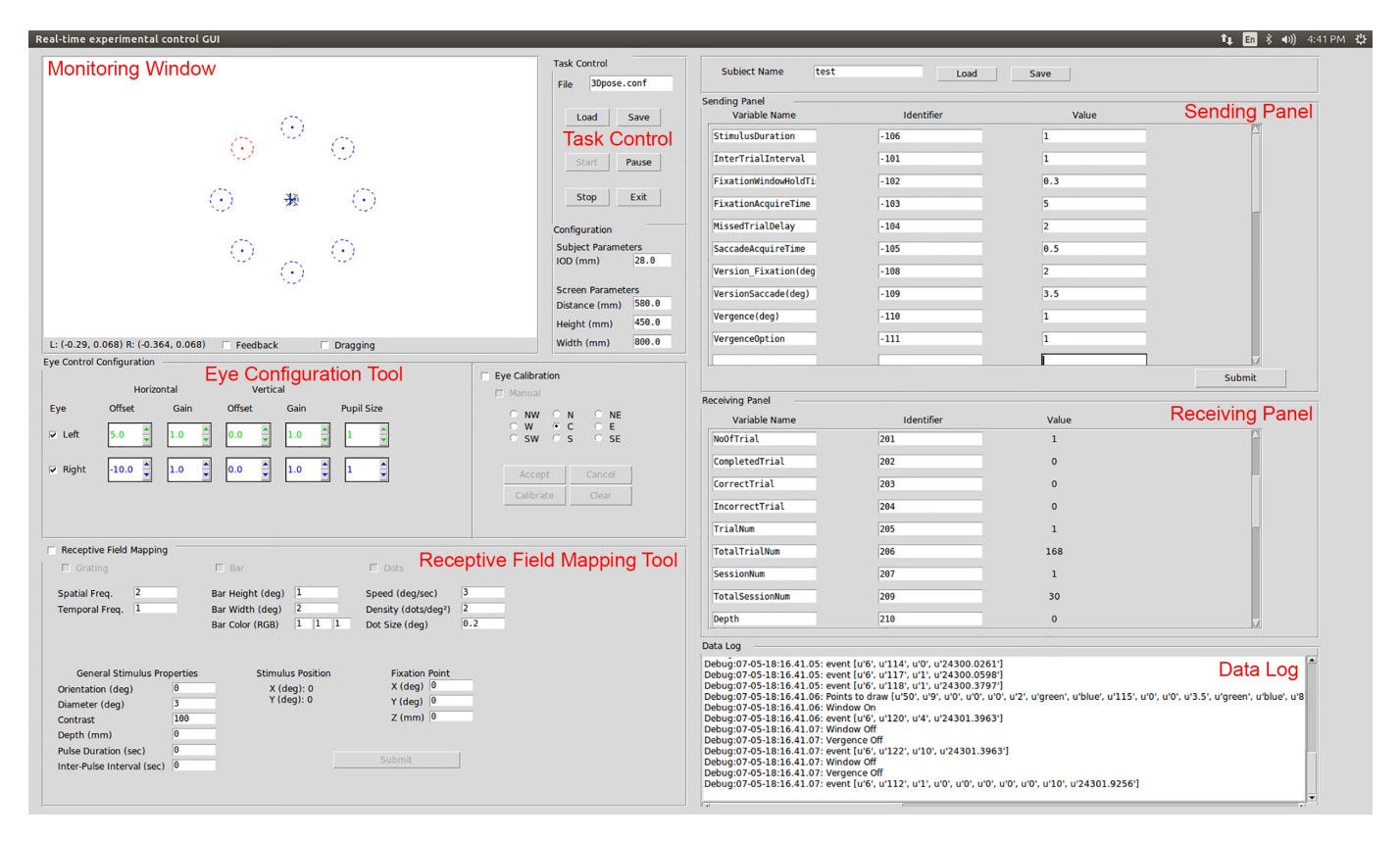

**Figure 4.** Graphical user interface. The GUI is a customizable experimental control panel, configured here for vision experiments. The upper left corner is a monitoring window, showing a scaled depiction of the visual display. Fixation windows and eye position markers are at the center. Eight choice windows for Application 1 are also shown (correct choice in red, distractors in blue). Below the monitoring window are eye configuration and receptive field mapping tools, which can be substituted for other experiment-specific tools. Task control for starting, pausing, or stopping a protocol is at the center top, along with subject- and system-specific configuration parameters. The sending panel in the upper right allows the experimenter to modify task parameters in real time. The receiving panel below that is used to display information about the current stimulus and experiment progress. The lower right panel shows a data log.

DOI: https://doi.org/10.7554/eLife.40231.007

The following figure supplement is available for figure 4:

**Figure supplement 1.** GUI configured with programmable command buttons for Application 3.

DOI: https://doi.org/10.7554/eLife.40231.008

Information about the ongoing status of an experiment and graphical feedback of critical data is often required to monitor and adjust parameters in real time. The GUI displays continuously sampled measurements (e.g., eye position, animal location, joint configuration, firing rate, etc.) in the monitoring window. To visually evaluate contingencies, the monitoring window can display boundary conditions (e.g., fixation windows, or demarcated areas of an arena). Such boundaries can be used to automatically trigger event signals (e.g., TTL pulses) that control external devices such as a solenoid for delivering a liquid reward or electrical stimulator. An experimenter can turn boundaries on/off, change their size, number, locations, etc. in real time through inputs to the GUI (see user manual).

Such changes are implemented in the REC-GUI framework using UDP communications to send/receive strings (*Figure 3*, *Figure 3—figure supplement 1*). Each string in the UDP packet consists an identifier (e.g., −106; a number which the stimulus code uniquely associates with a specific variable such as 'StimulusDuration') and a value (e.g., one to specify a stimulus duration of 1 s), followed by a terminator (qqqq….q/padded to 1024 characters). In this example, the string would be −106 one qqqq….q/ (see user manual for details). The GUI contains separate panels for sending and receiving UDP packets between the experimental control CPU and CPUs implementing major task

groups (*Figure 4*). For example, in the sending panel, an experimenter can enter values (e.g., 1) for predefined variables (e.g., 'StimulusDuration') with associated identifiers (e.g., '−106') and click 'Submit' to send UDP packets to a stimulus CPU. In the first two applications, the sending panel sends control information to a stimulus CPU. In the third application, it sends information to a video tracking module. It can send information to any machine capable of receiving UDP packets, as required for an experiment. The receiving panel allows the experimenter to predefine identifiers and variable names for receiving and displaying information from other CPUs (*Figure 4*). For example, the current trial number can be displayed using the identifier 205 and variable name 'TrialNum'. In this way, ongoing experimental information can be monitored in real time. This text-based approach provides a simple way to reconfigure the GUI to meet the demands of different experimental paradigms.

The GUI also contains placeholders in the lower left corner that allow experimenters to configure interfaces for specialized tools. We currently provide two default configurations. The first has interfaces to control eye calibration and receptive field mapping for vision studies. This version was used for Applications 1 and 2 (*Figure 4*). The second has programmable command buttons for executing user-defined callback routines, and was used for Application 3 (*Figure 4—figure supplement 1*). These interfaces can be substituted with other tools by editing the Python GUI code (see user manual).

## Results

### Network and system performance tests

The REC-GUI framework uses network-based parallel processing to implement experimental control and synchronize devices. In this section, we use Application 1 (neural basis of 3D vision in non-human primates) to test the latency of communication between the stimulus and experimental control CPUs resulting from hardware and software processing, as well as the performance of the main loop responsible for stimulus rendering and presentation. These tests demonstrate that the REC-GUI framework attains accurate and precise experimental control while implementing computationally demanding routines and enforcing behavioral contingencies.

Since different experimental processes ('major groups') are implemented on separate CPUs that communicate over a network, it is possible that limitations in network capacity can introduce delays that adversely affect performance. To test this possibility, we measured the delay resulting from the network hardware. Network latency was measured using a simple 'ping' command that is used to test reachability, signal fidelity, and latency in network communication between two hosts (*Abdou et al., 2017*). This ping sends an internet control message protocol requesting an echo reply from the target host. The latency of the ping is the round-trip duration of the packets between the two hosts. Network latency between the experimental control and stimulus CPUs was negligibly small (average latency = 24 µs; standard deviation: $\sigma$ = 5.1 µs; N = 1000 pings), indicating that the network hardware did not introduce substantial delays that could adversely affect performance (*Figure 5A*).

Next we measured the overall system performance which includes hardware delays tested above as well as software delays from the main loops (i.e., all executed processes) on both CPUs. In this test, the GUI sends a UDP packet to the stimulus CPU and after that packet is detected, MATLAB returns another UDP packet. We measured the latency of multiple round-trip sent/received packets (GUI → MATLAB → GUI), and the distribution of latencies is shown in *Figure 5B*. On average, the total duration of the round-trip packets for the first application was 6.79 ms ($\sigma$ = 2.9 ms; N = 500). This latency determines the time interval required to synchronize the CPUs when the experiment includes a computationally intensive task (in this case, rendering large 3D stimuli with depth variations and occlusion testing). For many experiments, performance would exceed this level. To determine an upper-bound of performance with this configuration, we removed all processes related to generating the 3D visual stimuli from the MATLAB main loop and measured the round-trip latency of the UDP packets. The average round-trip latency dropped to 4.3 ms ($\sigma$ = 2.3 ms; N = 500; *Figure 5C*). Note that this latency only limits real-time control, not the temporal alignment that can be achieved offline, as shown below using neuronal recording tests. Even with the longer latencies

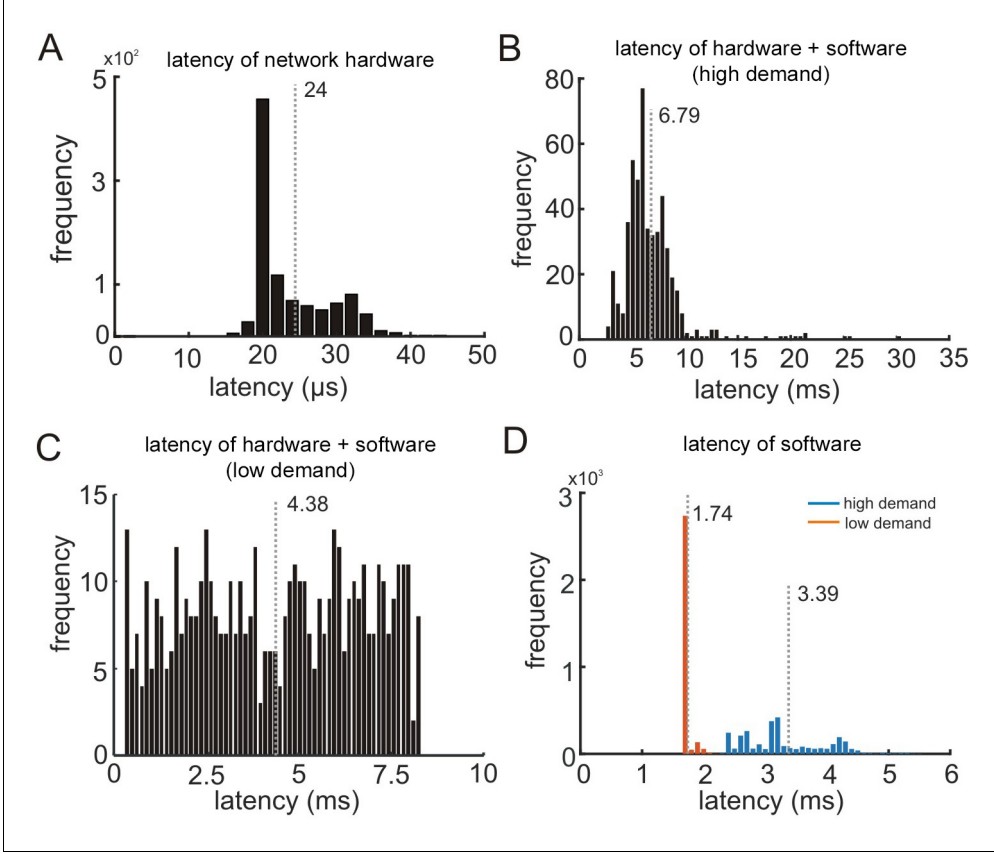

**Figure 5.** System performance. (**A**) Latency introduced by the network hardware (N = 1000 pings). (**B**) Overall system performance measured using the round-trip latency of UDP packets between the experimental control (GUI) and stimulus (MATLAB) CPUs with complex stimulus rendering routines ('high demand' on the system; N = 500 round-trip packet pairs). (**C**) Same as B without the stimulus rendering routines ('low demand' on the system; N = 500). (**D**) Duration of the main while-loop in the stimulus CPU with complex stimulus rendering routines (blue bars) or without (orange bars), N = 3000 iterations each. Vertical gray dotted lines mark mean durations.
DOI: https://doi.org/10.7554/eLife.40231.009

that occurred when rendering 3D stimuli, the achieved millisecond-level of control synchronization is sufficient for most experiments.

Lastly, we measured the duration of the main loop in the stimulus CPU, excluding the UDP processing time for communication with the experimental control CPU. This test includes all serially executed processes associated with stimulus rendering and presentation, as well as conditional statements for controlling experimental flow. The duration of the main loop was on average 3.39 ms ($\sigma$ = 1.3 ms; N = 3000 iterations) when rendering the 3D stimuli, thus accounting for about half the time required to synchronize the CPUs (blue bars in *Figure 5D*). To determine a lower-bound duration of the MATLAB main loop with this configuration, we removed the stimulus-related routines. After removing this code, the average latency of the main loop dropped to 1.74 ms ($\sigma$ = 0.11 ms; N = 3000 iterations; orange bars in *Figure 5D*). These tests demonstrate that the REC-GUI framework can facilitate complex experimental tasks in real-time, with very low latencies, and using high-level programming environments.

## External device control tests

The ability to accurately and precisely control external devices is critical for experimental studies. In this section, we use the first application (neural basis of 3D vision in non-human primates) to test the ability of the REC-GUI framework to implement such control over the real-time rendering and

presentation of large, computationally intensive stereoscopic stimuli at 240 Hz while enforcing fixation. As discussed in the Materials and methods, the concern is that this combination can cause dropped frames and/or time lags. Indeed, previous attempts to use high-level programming environments to present stimuli at this high frame rate while enforcing fixation have failed (see Discussion). To assess the fidelity of the presentation, phototransistor circuits were used to track the appearance of stimuli on the screen by detecting a small bright patch in the lower right or left corner of the corresponding eye's half-image (see user manual).

Voltage traces produced by the phototransistor circuits were saved by the Scout Processor to provide a precise signal for aligning events to the stimulus as well as to confirm the fidelity of stimulus presentation on a trial-by-trial basis. For example, if dropped frames occur on a certain trial, that can be detected and the trial discarded during data analysis. Representative traces showing the presentation of right (blue) and left (orange) eye images are shown in *Figure 6A*. The latency between the initial flip command and the appearance of the stimulus was very short (average delay = 4.71 ms, $\sigma$ = 0.16 ms, N = 500 trials), which is only ~ 540 µs longer than the video refresh cycle of 4.17 ms (*Figure 6B*). The fidelity of alternating right and left eye frames was confirmed by assessing the time difference between the two voltage traces. Since the stereoscopic images were presented at 240 Hz (120 Hz per eye), the right and left eye frame signals should be temporally shifted by ~ 4.17 ms. We measured the time difference between each alternation of the right and left eye frames over the 1 s stimulus duration for 500 trials. The average time difference was 4.16 ms ($\sigma$ = 0.066 ms; min = 3.9; max = 4.4), indicating that the right and left eye frames were well synchronized at the intended 240 Hz stimulus presentation rate (*Figure 6C*). This also confirms that no frames were dropped since dropped frames would appear as time differences $\geq$ 12.5 ms. These test results

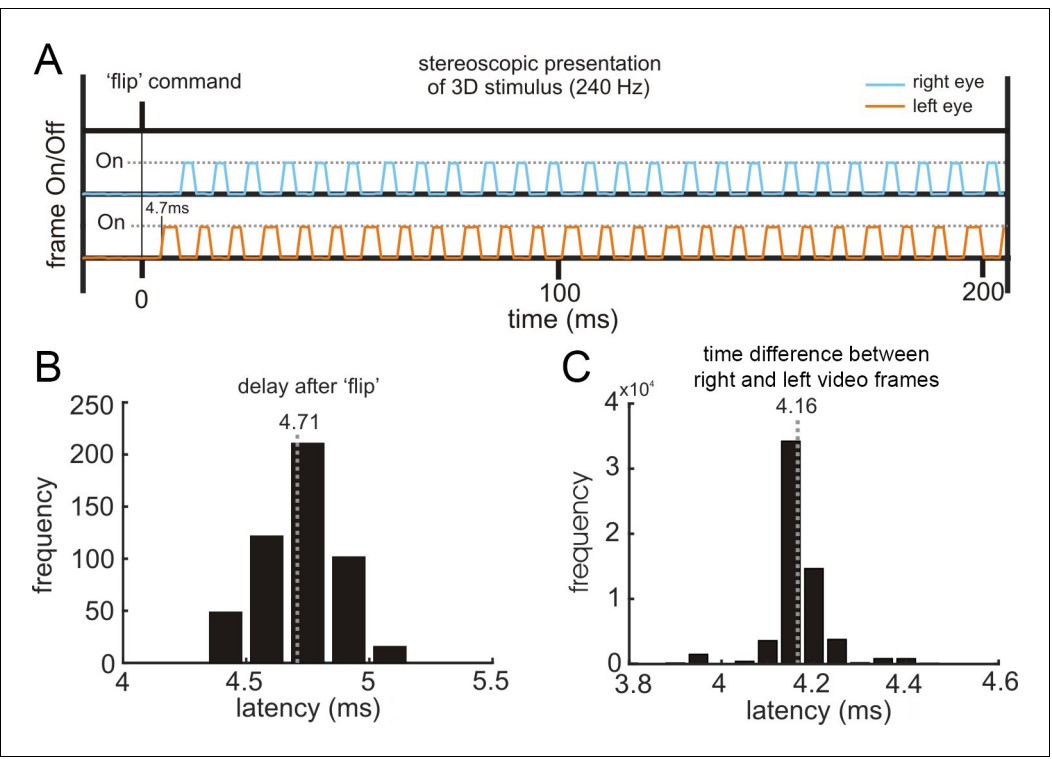

**Figure 6.** Quantifying the fidelity of external device control. (**A**) Right and left eye frame signals measured on the screen. The anti-phase rise and fall of the two signals indicates that they are temporally synchronized. (**B**) Latency between the initial flip command in MATLAB and the appearance of the stimulus (N = 500 trials). (**C**) Time differences between the two eyes' frames peak at 4.16 ms, indicating that the intended 240 Hz stimulus presentation was reliably achieved (N = 500 trials). Voltage traces were sampled at 30 kHz. Mean times are indicated by vertical gray dotted lines.

DOI: https://doi.org/10.7554/eLife.40231.010

demonstrate that the REC-GUI framework can use high-level programming environments to achieve accurate and precise control of external devices during computationally demanding tasks.

## Verification of real-time behavioral monitoring

Experimental control based on real-time monitoring of ongoing processes is critical for many studies. In this section, we confirm that the REC-GUI framework can be customized for a variety of neuroscience questions by presenting behavioral monitoring results from applications involving humans, non-human primates, and rodents.

*Application 1: Neural basis of 3D vision in non-human primates.* In this application, we presented 3D planar surfaces contingent on the monkey maintaining fixation on a target (*Figure 2A*). Eye position data were relayed to the GUI which implemented routines to evaluate if the monkey was holding fixation on the target, if fixation was broken, or if a particular choice was made. Depending on the results of these routines, the GUI directed the stimulus CPU to enter a particular experimental state (e.g., fixation only, stimulus on, choice targets on, etc.; *Figure 3—figure supplement 1*).

To demonstrate that the GUI successfully monitored the monkey's eye positions, we show representative horizontal and vertical eye displacements as a function of time for 34 successfully completed trials in *Figure 7A*. Two-dimensional (2D) eye traces showing that saccadic eye movements to each of the eight choice targets were accurately detected are shown in *Figure 7B*. As a further test, we implemented a fixation-in-depth task to show that the GUI can successfully enforce both version and vergence eye positions simultaneously. Fixation targets were presented at three depths relative to the screen, and the monkey was required to maintain fixation for 1 s. As shown in *Figure 7—figure supplement 1*, the GUI successfully enforced both version and vergence gaze contingencies. These tests demonstrate that the REC-GUI framework can implement experimental control contingent on real-time behavioral measurements.

*Application 2: 3D vision in humans.* Similar to the first application, we presented 3D planar surfaces contingent on the participant maintaining fixation on a target (*Figure 2B*). Representative horizontal and vertical eye displacements are shown as a function of time for 34 successfully completed trials in *Figure 8A*. The movements of the computer mouse used to select one of the sixteen choice targets are shown for the same trials in *Figure 8B*. The 2D traces of both eyes and the mouse cursor are shown *Figure 8C*. This test shows that the REC-GUI framework can implement experimental control contingent on inputs from multiple external devices (i.e., eye tracker and computer mouse).

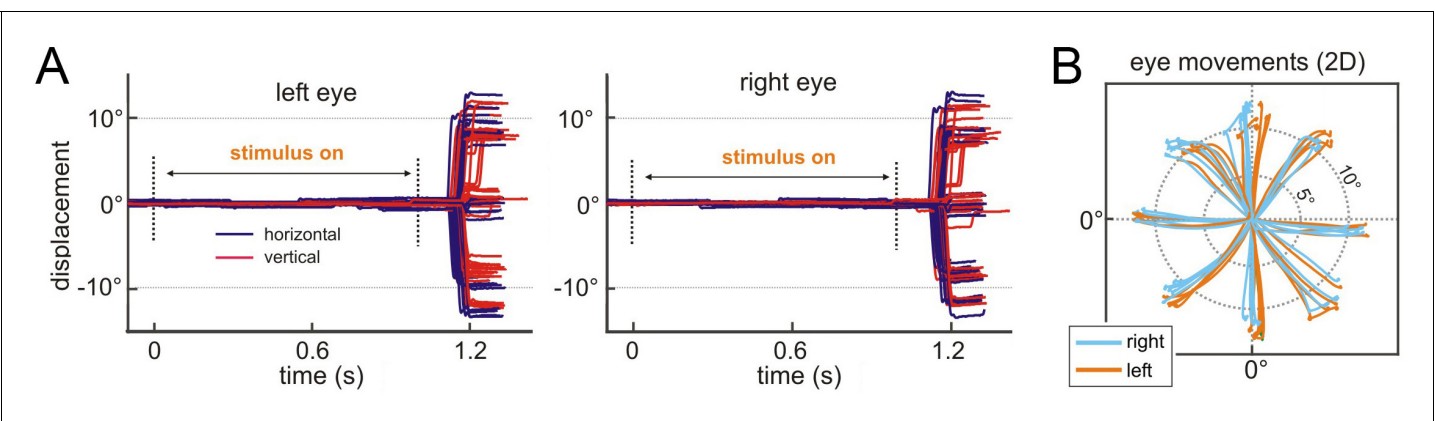

**Figure 7.** Application 1: Neural basis of 3D vision in non-human primates. Verifying behavior-contingent experimental control. (**A**) Left and right eye traces as a function time during the 3D orientation discrimination task, aligned to the stimulus onset (N = 34 trials). The traces end when the choice was detected by the GUI. Horizontal and vertical components of the eye movements are shown in purple and red, respectively. Version was enforced using a 2° window. Vergence was enforced using a 1° window (data not shown). (**B**) Two-dimensional (2D) eye movement traces shown from stimulus onset until the choice was detected by the GUI (same data as in **A**).

DOI: https://doi.org/10.7554/eLife.40231.011

The following figure supplement is available for figure 7:

**Figure supplement 1.** Enforcement of version and vergence during fixation at different depths.

DOI: https://doi.org/10.7554/eLife.40231.012

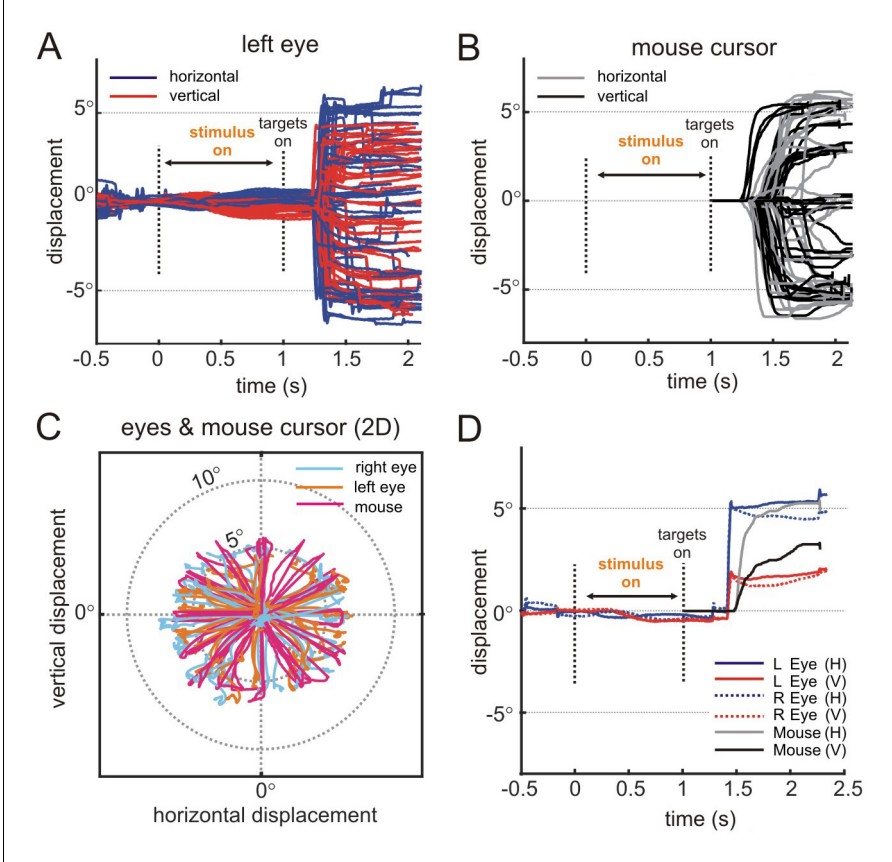

**Figure 8.** Application 2: 3D vision in humans. Verifying behavior-contingent experimental control. (**A**) Left eye traces as a function time during the 3D orientation discrimination task, aligned to the stimulus onset (N = 34 trials). The traces end when the choice (mouse click on a target) was detected by the GUI. Horizontal and vertical components of the eye movements are shown in purple and red, respectively. Version was enforced using a 3° window. Vergence was enforced using a 2° window (data not shown). (**B**) Computer mouse cursor traces for the same trials. The cursor appeared at the location of the fixation target at the end of the stimulus presentation. (**C**) 2D eye and mouse cursor traces ending when the choice was detected by the GUI. (**D**) Movement of both eyes and the mouse cursor for a single trial. Note that the movement of the mouse cursor was delayed relative to the movement of the eyes but caught up shortly after movement initiation.
DOI: https://doi.org/10.7554/eLife.40231.013

Since all data analyzed in this subsection were saved by the experimental control CPU, this test further shows that inputs from external devices can be precisely aligned to stimulus-related events without a dedicated data acquisition system, such as the one used in Application 1. For example, this makes it possible to analyze the coordination of different motor effectors, such as the eyes and arms, without a dedicated acquisition system. Along this line, note that when selecting the choice target, the movement of the computer mouse initially lagged behind the movement of the eyes (*Figure 8D*).

*Application 3: Passive avoidance task in mice.* We implemented a passive avoidance task in which a two-way shuttle box was divided into bright and dark rooms by a gate (*Figure 2C*). Position within the bright room was automatically monitored using a video tracking module. The mouse initially explored a large portion of the bright room, and promptly entered the dark room when the gate was opened (*Figure 9A*). After activating the shock scrambler, when the GUI detected that the mouse entered the dark room, it sent a control signal to the scrambler to deliver a small electric shock (<0.9 mA). The entry door was manually closed. The mouse was removed from the dark room after ~ 3 s of shock delivery, and placed in its home cage for ~ 5 min. The mouse was then returned to the bright room (*Figure 9B*). After the foot shock, the mouse explored a smaller area of the bright

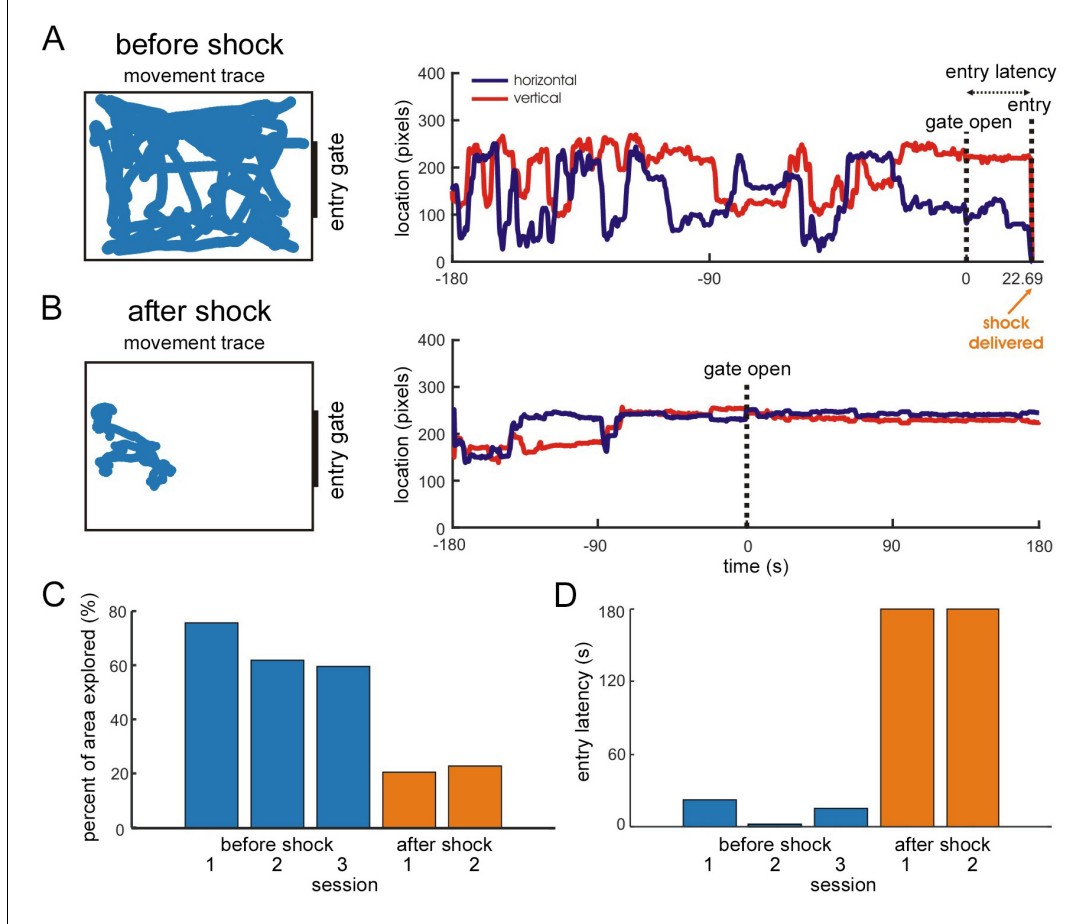

**Figure 9.** Application 3: Passive avoidance task in mice. Verifying automated behavior-contingent experimental control. (**A,B**) The left column shows the movement of a mouse (blue trace) within the bright room before (**A**) and after (**B**) foot shock. The right column shows the movement as a function of time, and marks the gate opening. In *A*, the mouse's entry into the dark room and the foot shock are also marked. (**A**) Before foot shock. (**B**) After foot shock. (**C**) The percentage of the bright room explored by the mouse before (blue bars) and after (orange bars) foot shock. The mouse explored much less area after the shock. (**D**) Time to enter the dark room after the gate was opened. Before the foot shock (blue bars), the mouse quickly entered the dark room. After the foot shock (orange bars), the mouse never entered the dark room (sessions were ended after 3 minutes).
DOI: https://doi.org/10.7554/eLife.40231.014

room (*Figure 9C*) and never entered the dark room (*Figure 9D*). The entry latencies and movements of the mouse were saved on the experimental control CPU. This application provides an example of automating behavioral training and assessment using the REC-GUI framework. More broadly, this test shows that the REC-GUI framework can automate experimental control by transforming external device inputs (e.g., video tracking data) into control outputs for other devices (e.g., a shock scrambler).

## High precision temporal alignment of multiple data streams

We used the first application (neural basis of 3D vision in non-human primates) to confirm the ability to align neuronal, stimulus, and behavioral events in time. While the monkey performed the discrimination task, we measured the 3D surface orientation tuning of a CIP neuron. First, we confirmed the ability to precisely align stimulus-driven neuronal responses to the stimulus onset detected by the phototransistor. A raster plot showing the timing of action potentials for 545 trials aligned to the stimulus onset (each row is a different trial) along with the spike density function (red curve; convolution with a double exponential) is shown in *Figure 10A*. The alignment of spike times relative to the stimulus onset reveals a sharp visual transient, consistent with the properties of CIP. Second, we confirmed the ability to align spike times to the choice saccades. Saccade onsets were detected offline

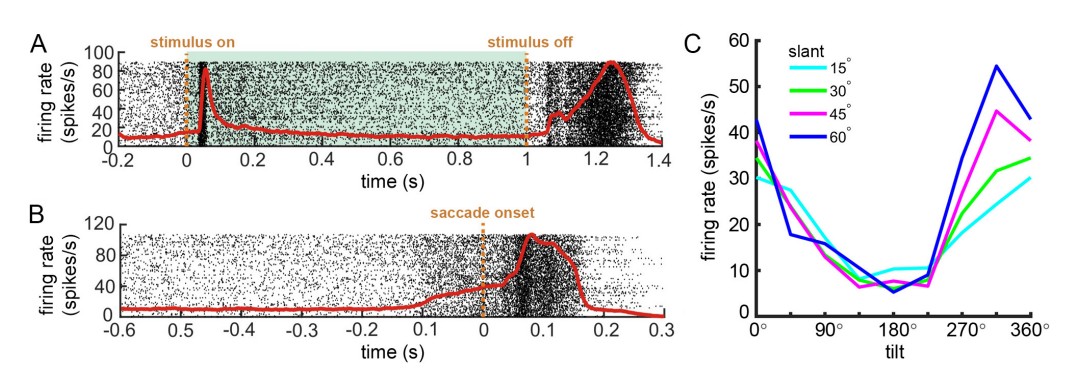

**Figure 10.** Temporal alignment of action potentials to stimulus-related and behavioral events. (**A**) Raster plot showing spike times aligned to the stimulus onset (N = 545 trials). Shaded region marks the stimulus duration. (**B**) Raster plot showing spike times aligned to the saccade onset. Each row is a different trial, and each dot marks a single action potential. Red curves are spike density functions. (**C**) 3D surface orientation tuning. Each curve shows tilt tuning at a fixed slant.

DOI: https://doi.org/10.7554/eLife.40231.015

as the first time point at which the eye movement was faster than 150°/s (*Kim and Basso, 2008*; *Kim and Basso, 2010*). Spike times were aligned to the saccade onset and the spike density function calculated (*Figure 10B*). Note the build-up of activity preceding the saccade that was not evident when the spike times were aligned to the stimulus onset. As expected in CIP (*Rosenberg et al., 2013*), the neuron was tuned for slant and tilt (*Figure 10C*). These tests confirm the ability to precisely align events recorded using the REC-GUI framework.

## Discussion

Test results from applications involving humans, non-human primates, and rodents confirm that the REC-GUI framework provides a solution for implementing demanding neuroscience studies with accurate and precise experimental control. By achieving millisecond-level control with high-level programming environments, the framework can help researchers overcome technical challenges that hinder research, without the need for low-level programing languages or professional programmers. Beyond experiments with a fixed trial structure, the framework can support research involving the use of naturalistic, complex stimuli that are dynamically updated based on real-time behavioral or neural measurements. In Application 3, we showed that the framework can automate real-time behavioral monitoring for tasks without a fixed trial structure, and perform experimental manipulations based on defined criteria. Similarly, the framework can support more complex closed-loop experiments in which multisensory visual–vestibular stimuli are updated based on active steering behaviors rather than passive, predefined motion profiles. Alternatively, stimuli can be updated or neural activity perturbed using short-latency stimulation triggered by real-time behavioral or neural events. Such capabilities will be critical to understanding the relationship between the dynamic activity of neural populations, perception, and action during natural behaviors. The framework can facilitate such research by reducing the overhead associated with parlaying the necessary technology into a cohesive control system. Thus, a major contribution of the REC-GUI framework is in achieving experimental flexibility and high temporal precision while minimizing coding demands. While it is inescapable that programming is required for customization, the level of coding ability possessed by many graduate students (e.g., translating the logical steps to move through a series of experimental states into MATLAB or Python code) is sufficient to implement diverse applications. Indeed, by reducing technical hurdles and providing a flexible experimental framework, complex protocols will be accessible to a greater number of labs, and the opportunity for discovery increased.

### Advantages and limitations of the REC-GUI framework

The REC-GUI framework offers several advantages compared to other currently available control systems. First, network-based parallel processing makes the framework inherently modular and highly

flexible. Since network-based parallel processing is a standard approach to coordinating multiple CPUs, it is well-documented online, and therefore setting up the necessary networks does not pose a substantial technical hurdle. Moreover, since the only constraint on incorporating specialized software or hardware is that network interfacing is supported, a broad range of devices for stimulus presentation (visual displays, speakers, pellet droppers, etc.), behavioral measurement (button presses, eye movements, position, etc.), and other experimental needs (stimulator, osmotic pump, etc.) can be readily incorporated into experimental setups. Such flexibility allows the framework to be adapted to a broad range of preparations (*in vitro*, anesthetized, or awake-behaving) and research domains (sensory, cognitive, motor, etc.), and makes it agnostic to the type of data recorded (electrophysiological, optical, magnetic resonance imaging, etc.). Second, dividing computing demands across CPUs improves system performance and enables researchers to implement system components using different coding languages and operating systems. This reduces compatibility issues and increases the efficiency with which experimental setups are configured. For example, some applications require multiple external acquisition systems. While we have not tested the framework with such an application, it would be straightforward to temporally align events saved on the different systems by having the GUI send synchronizing signals to each of them. Likewise, while the performance of the REC-GUI framework, as implemented here, is sufficient for most experimental needs, if greater computational speed is required, that can be achieved using a lower-level programming language such as C or C ++ for stimulus rendering and presentation. Even greater processing speed could be achieved using an onboard microprocessor with software loaded onto the ROM. While this is possible with the REC-GUI framework, such implementations would increase the coding demands, defeating the goal to minimize coding efforts. Third, diverse experimental components (behavior, neural recordings, stimulation, etc.) can occur in parallel at different temporal resolutions, and be precisely aligned, allowing for multi-faceted research. Lastly, a user-friendly, fully customizable GUI allows for intuitive and flexible experimental design changes.

## Comparisons with other control systems

We briefly benchmark several existing control systems against the REC-GUI framework. One commercially available system that is used in a broad range of neuroscience applications is Spike2 (CED, Inc.). That system includes software and multifunctional hardware for analog and digital interfacing (including multiple analog-to-digital channels with large bandwidths for neural recordings), providing robust experimental control and data acquisition for experiments such as *in vitro* patch clamp recordings (*Hasenstaub et al., 2005*), recordings of stimulus-driven neuronal activity in anesthetized cats (*Rosenberg and Issa, 2011*), and multisensory studies in awake-behaving monkeys (*Rosenberg and Angelaki, 2014a*). One potential downside is that the Spike2 scripting language is designed for their acquisition hardware, and sufficiently complex that the company hosts training events. In contrast, the REC-GUI framework is not coupled to specific hardware, and relies on widely used high-level programming environments.

The Laboratory of Sensorimotor Research (LSR) real-time software suite is a free package including a GUI, stimulus rendering, data acquisition, and offline analysis tools (*Hays et al., 1982*). Similar to REC-GUI, the LSR suite divides experimental control, stimulus processing, and data acquisition across CPUs. For the LSR suite, this is achieved using TCP or direct digital connections with National Instruments Cards. The LSR suite is highly powerful, and can meet the demands of complex behavioral paradigms. However, the LSR scripting language is complex, making the overhead of learning the system and developing new experiments quite high (*Eastman and Huk, 2012*). The REC-GUI framework reduces this overhead, particularly as it relates to modifying or adding new experimental tasks, by achieving accurate and precise experimental control with high-level programming environments that are widely used by the neuroscience community. Additionally, the LSR suite is largely specialized for visuomotor studies, whereas the REC-GUI framework is agnostic to the area of neuroscience research.

A freely available system that is fully implemented in MATLAB is MonkeyLogic (*Asaad et al., 2013*). The system achieves millisecond-level temporal resolution, and provides a user interface with real-time behavioral monitoring. In addition, it is convenient to implement control flows for new behavioral tasks. MonkeyLogic is designed for a single CPU, so experimental control is performed serially due to MATLAB's multithreading limitations. This can be problematic for real-time control when the stimulus demands are high. For example, since stimuli are transferred to the video buffer

during the inter-trial interval without compression, it may not be suitable for presenting long-duration stimuli. Along this line, the MonkeyLogic forum indicates that it cannot support 240 Hz visual stimulus presentation, as used in the first application we tested (http://forums.monkeylogic.org/post/high-refresh-rates-vpixx-8408242). Furthermore, this system is limited to experiments with well-defined trial structures, rather than closed-loop or continuous experiments (e.g., Application 3 described here). Display limitations will exist for other single CPU, MATLAB-based control systems, though some limitations may be partially remediated if the real-time monitoring and control features provided by a GUI are eliminated (*Eastman and Huk, 2012*). Such systems may be ideal for tasks that do not have high real-time behavioral contingency and stimulus demands, since network communications make the REC-GUI framework more complex. There is also some added cost compared to single CPU systems since REC-GUI requires multiple CPUs. In exchange, the REC-GUI framework allows high-level programming environments to be used to robustly control computationally demanding and behaviorally complex experiments.

## Ongoing open-source development

We will maintain a webpage for supporting the REC-GUI framework as an open-source project (https://recgui2018.wixsite.com/rec-gui). The webpage currently includes a forum for discussion and assistance with customization, as well as links to relevant downloads. To track versioning associated with future developments and customization, the software is stored on GitHub (*Kim et al., 2019*; copy archived at https://github.com/elifesciences-publications/rec-gui). To increase the generality and functionality of the framework, we encourage others to make development contributions. Our hope is that REC-GUI will help researchers perform cutting-edge research by reducing time spent solving technical problems and increasing time focused on experimental questions and design. The REC-GUI framework can also help promote research transparency, standardize data acquisition, and improve reproducibility by facilitating the replication of experimental paradigms.

## Acknowledgments

We thank Heather Mitchell, Xinyu Zhao, Qiang Chang, and the IDD Models Core supported by Waisman Center IDDRC (U54HD090256) for help with the passive avoidance task. This work was supported by the Alfred P. Sloan Foundation, Whitehall Foundation Research Grant 2016-08-18, Shaw Scientist Award from the Greater Milwaukee Foundation, and National Institutes of Health Grants DC014305 and EY029438. Further support was provided by National Institutes of Health Grant P51OD011106 to the Wisconsin National Primate Research Center, University of Wisconsin – Madison.

## Additional information

### Funding

| Funder | Grant reference number | Author |
|---|---|---|
| Alfred P. Sloan Foundation | FG-2016-6468 | Ari Rosenberg |
| Whitehall Foundation | 2016-08-18 | Ari Rosenberg |
| National Institutes of Health | DC014305 | Ari Rosenberg |
| National Institutes of Health | EY029438 | Ari Rosenberg |
| Greater Milwaukee Foundation | Shaw Scientist Award | Ari Rosenberg |

The funders had no role in study design, data collection and interpretation, or the decision to submit the work for publication.

### Author contributions

Byounghoon Kim, Conceptualization, Resources, Data curation, Software, Formal analysis, Validation, Investigation, Visualization, Methodology, Writing—original draft, Writing—review and editing; Shobha Channabasappa Kenchappa, Software, Methodology, Writing—original draft; Adhira

Sunkara, Conceptualization, Software, Methodology, Writing—original draft, Writing—review and editing; Ting-Yu Chang, Software, Validation, Investigation, Methodology; Lowell Thompson, Software, Validation, Visualization, Writing—original draft; Raymond Doudlah, Resources, Validation, Visualization; Ari Rosenberg, Conceptualization, Resources, Data curation, Software, Formal analysis, Supervision, Funding acquisition, Validation, Investigation, Visualization, Methodology, Writing—original draft, Project administration, Writing—review and editing

### Author ORCIDs
Byounghoon Kim (iD) http://orcid.org/0000-0001-7159-5134
Ting-Yu Chang (iD) http://orcid.org/0000-0003-3964-0905
Ari Rosenberg (iD) http://orcid.org/0000-0002-8606-2987

### Ethics
Human subjects: This study was performed in strict accordance with the Declaration of Helsinki. All experimental procedures were approved by the Institutional Review Board at the University of Wisconsin-Madison (Protocol #: 2016-1283). All participants provided informed consent.
Animal experimentation: This study was performed in strict accordance with the recommendations of the National Institutes of Health's Guide for the Care and Use of Laboratory Animals. All experimental procedures and surgeries were approved by the Institutional Animal Care and Use Committee (IACUC) at the University of Wisconsin-Madison (Protocol #s: G005229, G5373).

### Decision letter and Author response
Decision letter https://doi.org/10.7554/eLife.40231.018
Author response https://doi.org/10.7554/eLife.40231.019

## Additional files
### Supplementary files
• Transparent reporting form
DOI: https://doi.org/10.7554/eLife.40231.016

### Data availability
All source code for implementing the Real-Time Experimental Control with Graphical User Interface (REC-GUI) framework is available for download through the REC-GUI GitHub: https://github.com/rec-gui/rec-gui (copy archived at https://github.com/elifesciences-publications/rec-gui).

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
