## [Decision Letter]

[**Editorial note:** This article has been through an editorial process in which the authors decide how to respond to the issues raised during peer review. The Reviewing Editor's assessment is that all the issues have been addressed.]

Thank you for submitting your article "Real-time experimental control using network-based parallel processing" for consideration by *eLife*. Your article has been reviewed by three peer reviewers, including Sacha B Nelson as the Reviewing Editor and Reviewer #1, and the evaluation has been overseen by Michael Frank as the Senior Editor. The following individual involved in review of your submission has also agreed to reveal their identity: Niraj Desai (Reviewer #2).

We strongly encourage you to revise your manuscript to deal with the issues raised in the reviews. As you can see, the reviewers feel that the present version does not live up to the claims you make for it, and we hope that you revise it to come far closer to a more useful contribution. As you know, these reviews and your responses will be published with the manuscript, as well as the editors' assessment of how responsive your revision is to the reviews.

Summary:

This is a description of a software package intended to generalize the problem of real time acquisition and control of rapidly changing signals, such as might be encountered in neuroscience experiments relating neurobiological signals to behavior. Although this is a potentially highly useful undertaking, the paper describes only a single use case involving primate vision experiments and many of the features of the described software seem highly specialized for this application with little information about the true generality of the described tool. In addition, reviewers were concerned that the overall approach lacked novelty and so a mere "proof of principle" was of limited value.

Major concerns:

As you can see in the reviews below, the reviewers largely agree on these central recommendations:

1) Maximize the generality of the described tool/resource by demonstrating application to multiple diverse experimental settings and by providing information on how to customize the software for any of a range of applications. This could be achieved, for example, by briefly describing the application to the psychophysics experiments alluded to, and by collaborating with another lab to tailor the system to their acquisition/control needs in a very different (ideally non-vision) setting.

2) Describe the performance in more general terms (e.g. as is done in Figure 5), rather than in terms specific to the major use case for which it was developed. This also applies to the discussion, which is now very focussed on software for monkey vision experiments.

3) Think through and describe the resources needed to support application of the software to other use cases. This includes a plan for maintenance of the software (e.g. via GitHub) as well as much more detailed supplementary instructions for modification and use.

4) Clearly state existing limitations to the generality. State what has and has not been tested and what is and is not possible with the existing software.

5) Although the issue of novelty of the approach was raised by one of the reviewers, and this view is shared, it will likely not be helpful to try to rewrite the manuscript to stress the novelty. Taking steps to ensure the usefulness and then highlighting them will probably be more fruitful in terms of improving the manuscript.

Separate reviews:

*Reviewer #1:*

The authors describe a new software suite for data acquisition in neurophysiology experiments. The intriguing features promised in the early parts of the paper are that it is highly modular, agnostic to coding language and operating system, and hence broadly applicable to many types of real time data acquisition and control experiments. The authors achieve high performance in terms of simultaneously computing stimuli, monitoring behavior and acquiring neurophysiological signals by segregating tasks and running them on separate processors linked by internet protocols, augmented by standard digital IO.

The rest of the paper is somewhat disappointing in that it mainly describes applicability to a single type of experiment (visual stimulation and recording of eye position and extracellular action potentials in behaving primates). Hence it is not likely to be obvious to most readers not performing very similar experiments, how they might configure the system for very different experiments. Ideally, a paper describing a general purpose system would apply it to more than one system. The figures are mostly devoted to the performance relevant to their specific experiment and so will not interest many readers. Figure 5 gives some idea of the performance in a general sense of the interaction between components, but Figures 3,4,6,7,8 are highly specific to the particular project and so are really not especially relevant to most readers interested in a more general tool. These figures would mostly be better as supplements to figures that measure performance or describe capabilities in a more general way. This lack of generality is exacerbated by the fact that the link to the downloads related to the software is nonfunctional.

Readers would want to know things like: which operating systems were actually tested? Which coding languages were used? Which acquisition hardware was tested? What aspects of the GUI are configurable and which are hard coded? They would want examples of experiments performed that are of multiple types involving different types of signals and different control paradigms. Otherwise, this publication would really be much more appropriate for a more specialized audience (e.g. the monkeyLogic paper was published in J. Neurophysiol. the Psychophysics Toolbox paper was in Spatial Vision and an update was published in Perception).

Finally, the discussion contains comparisons to other systems, but these comparisons are again formulated in terms of a very specific experiment's requirements (e.g. 240 Hz visual stimulus presentation) rather than in more generic terms of bandwidth for acquisition and control.

Additional data files and statistical comments:

The support for the software is a critical part of publication and of evaluation of the manuscript. The web site allowing access to the User Manual and other downloads was not accessible, even though these were included with the submission. As noted above, a User Manual for modifying the software and applying it to other kinds of experiments is also needed.

*Reviewer #2:*

Kim and colleagues present a parallel processing framework to integrate electrophysiological (or other neural) measurements, stimulus generation, and behavioral control. Their basic idea is that complex experiments can be run in a flexible manner if different tasks are spread out between different CPUs, which communicate with each other via standard internet protocols (TCP and UDP). The idea is intriguing and the implementation the authors themselves use is an excellent proof of principle. The major weakness is that the authors offer very little guidance to researchers who have different equipment or study a different experimental system, or simply have different experimental objectives and constraints. The project as it stands – the manuscript, the supplementary software examples, and the (under construction) website – is too particular to the authors' own work; it would be difficult for others to generalize from this.

Strengths:

1) The basic idea is clever: breaking up complex experiments into modules definitely seems like a good way to combine flexibility, computationally-intensive processes, and temporal fidelity. And exploiting existing protocols for network communication should make integration of these processes seamless. Many types of experimental programs could benefit from the general approach.

2) The kinds of experiments these authors themselves do – awake monkeys, complex visual stimuli, real-time control – are very technically challenging. Their success in using the general approach in this particular case is therefore reassuring, and their software likely will help others who do the same kinds of experiments.

Weakness:

3) But what about everybody else? The authors make broad claims for their framework:

"Since the only constraint on incorporating specialized software or hardware is that network interfacing is supported, a broad range of devices for stimulus presentation (visual displays, speakers, pellet droppers, etc.), behavioral measurement (button presses, eye movements, biometrics, location, etc.), and other experimental needs (stimulator, osmotic pump, etc.) can be used out of the box. Such flexibility allows the framework to be adapted to a broad range of experimental preparations (in vitro, anesthetized, or awake-behaving) and neuroscience research domains (sensory, cognitive, motor, etc.), and makes it agnostic to the type of neural data recorded (electrophysiological, optical, magnetic resonance imaging, etc.)."

I suspect these claims are true, but I also think that the authors haven't demonstrated that they're true. Their vision experiment is the only implementation described and the only example given. In fact, the authors' separate work on "perceptual learning studies with adolescents with autism" is alluded to, but not described.

More importantly, the authors never really explain – in detail – how to implement the framework. The descriptions are either too particular to the authors' own equipment or too general and/or vague to be useful. The manuscript assures readers that implementation is straightforward in language that is, perhaps, meant to be reassuring but comes off as glib: "Other acquisition systems that support network interfacing can be substituted with minimal effort.… The default GUI contains interfaces to control eye calibration and receptive field mapping, but these can be easily substituted with other tools." A reasonable reader response: "Okay.… sure.… if you say so.… but how do I do this?!"

In making this criticism, I'm motivated by the question of what a methods paper is for. My view is that the best methods papers offer not only a proof of principle but serve as a how-to guide. In the Introduction, the authors indicate that they hope their framework will benefit labs that lack the technical skills to design complex experimental systems themselves. That is a valuable goal – and a feasible one – but, in my opinion, they haven't reached that goal here. In fact, even labs with a lot of technical expertise but different equipment will need to "reinvent the wheel" to get this working.

What I would suggest is that the authors describe – perhaps in supplementary material so as not to disrupt the flow of the main text too much – how they built their system step by step by step. How both physically and in software do you connect a piece of hardware with a CPU using TCP? How do you set up a network interface card? Maybe show screen captures or pictures of the equipment. Do something to make this more tangible.

A related point: the software examples are not enough. Reading code written by someone else is one of life's great miseries, no matter the talent of the original programmer, and in this case is not especially helpful in understanding how the REC-GUI framework works.

Minor Comments:

1) The Introduction promises something very broad – a general system that can be used easily without great technical investment. Unless and until the rest of the manuscript fulfills this promises, it should be toned down. It should state more clearly and narrowly what is actually in the paper.

2) The project should live on Github or someplace similar. One reason open-source projects fail to gain traction is when they are seen as too proprietary. REC-GUI will only be important if it is adopted by a wide community, outside of the originating lab. The software shouldn't live on that lab's website. Also, the authors write: "We will maintain the REC-GUI framework as an open-source project.…." What does that mean exactly? Maintain it for a year? For five years? 10? 20? Things change in the life of a lab; people graduate and move on. If this project is to be real, it can't be tied to one lab.

3) Although the Abstract and Introduction talk about being independent of a particular operating system, but the website indicates that the Ubuntu version of Linux is "highly recommended" for the GUI. That is a real limitation and should be mentioned explicitly in the manuscript.

*Reviewer #3:*

The authors describe a UDP/TCP approach for coordinating eye tracking, electrophysiology, and behavioral input across multiple computers. The loop interval is about 5ms. The authors can transmit information among the various CPUs, and presumably more CPUs could be integrated into this system. The authors have done a good job of writing down and describing their system. It is faster than most, with a 5ms cycle time.

My biggest problem with the article is that there really isn't any novelty. Every lab I've been in has solved this problem similarly, with multiple CPUs and some communication across them via the network. Some labs I know use some combination of UDP/TCP, file systems, and digital TTL signals. In my opinion, this article doesn't belong in *eLife*, I don't think it will appeal to its broad readership. It's not novel enough.

At the same time, not to discourage the authors, I applaud the fact that they have written this down and have made it available to other groups.

---

## [Author Response]

Major concerns:As you can see in the reviews below, the reviewers largely agree on these central recommendations:1) Maximize the generality of the described tool/resource by demonstrating application to multiple diverse experimental settings and by providing information on how to customize the software for any of a range of applications. This could be achieved, for example, by briefly describing the application to the psychophysics experiments alluded to, and by collaborating with another lab to tailor the system to their acquisition/control needs in a very different (ideally non-vision) setting.

Based on the reviewers’ comments and a follow-up discussion with the editor, our revisions focus on demonstrating the generality of the REC-GUI framework using three distinct applications with humans, non-human primates, and rodents. The non-human primate application confirms the ability of the framework to accurately and precisely control external hardware under computationally demanding scenarios while achieving millisecond level precision. We added details of a human psychophysics application to show that the framework can coordinate experimental control based on multiple external device inputs. This application further shows that measurements made by multiple external devices can be saved and precisely aligned without external data acquisition systems. We added a passive avoidance task with mice to show how the framework can be used to automate behavioral training and assessment by transforming input device signals into control signals for external devices. Together, these three applications show generality across animal models, as well as experimental modalities (e.g., neural recordings during behavior, human psychophysics, and animal behavioral training and assessment).

We greatly expanded the user manual and supporting code. These provide detailed examples of coding structures used by the REC-GUI framework, including three ready-to-use scripts for implementing specific protocols: (i) a gaze fixation task (fixation.m), (ii) an eye calibration routine (calibration.m), and (iii) a receptive field mapping routine (receptive_field_mapping.m). The examples implement vision-related tasks, but the structures are applicable to a broad range of experiments, regardless of the sensory or motor modality investigated. For example, the eye calibration routine can be modified into a motor reach calibration routine with minimal effort. A main point of these examples is to illustrate how the GUI communicates with hardware and other software to implement tasks. We also include a basic template for implementing new protocols (start_coding.m), and code implementing a simple closed-loop task (start_coding_closedloop.m). In particular, the closed-loop task is a useful starting point for learning to use and customize the REC-GUI framework since it does not require specialized hardware or software (beyond Python and MATLAB).

We believe that these additions provide sufficient information to guide a wide range of customizations of the REC-GUI framework. We further emphasized that support will be provided through the REC-GUI forum (https://recgui2018.wixsite.com/rec-gui/forum) to cover any questions that may arise, and that the REC-GUI framework is an open-source project that will continue to evolve. We will routinely update the user manual based on developments and forum discussions. The user manual can be downloaded from the REC-GUI GitHub (https://github.com/rec-gui).

2) Describe the performance in more general terms (e.g. as is done in Figure 5), rather than in terms specific to the major use case for which it was developed. This also applies to the discussion, which is now very focused on software for monkey vision experiments.

We expanded the analyses quantifying system performance (Figure 5). Specifically, we characterize upper and lower bound estimates of performance when relying entirely on high-level programming environments to implement an experiment. We also provide specific test results from three applications using different system configurations to support studies with humans, non-human primates, and rodents.

To ensure that the testing results are relevant to experimentalists and readily interpretable, they were performed using specific applications. To increase the generality of the tests, we framed the motivations for them and the interpretation of the results in terms of broader experimental needs.

Within the Discussion, we expanded text regarding how the REC-GUI framework can be applied to a broad range of neuroscience questions ranging from in vitro experiments to closed-loop multisensory studies. In addition to other control systems previously discussed, we now discuss the Spike2 (CED, Inc.) system which is used in a wide range of applications.

3) Think through and describe the resources needed to support application of the software to other use cases. This includes a plan for maintenance of the software (e.g. via GitHub) as well as much more detailed supplementary instructions for modification and use.

The REC-GUI webpage (https://recgui2018.wixsite.com/rec-gui) hosts a discussion forum and contains links to essential downloads. In addition to the user manual, we indicate in the text that the forum is an important resource for customization. The added GitHub stores the user manual, software, and tracks versioning (https://github.com/rec-gui). Throughout the text, we provide examples of what would be needed for the framework to support applications that were not tested. These additions largely occur in the Materials and methods and Discussion sections. As discussed above, we expanded the user manual and included additional example code to further clarify modification and use.

4) Clearly state existing limitations to the generality. State what has and has not been tested and what is and is not possible with the existing software.

We have clarified limitations to the generality of the REC-GUI framework throughout the manuscript as they relate to hardware, software, and the types of experimental paradigms. For example, we emphasize that customization does require programming, but highlight that the necessary coding abilities are of the level expected of graduate students. We also reiterate that network interfacing is a requirement to integrate hardware or software with the framework. In the Discussion, we mention applications and extensions that have not been tested, and what would be required to implement them. We additionally discuss performance limitations, and how future developments can reduce those limitations (though in some cases, doing so would require more advanced programming abilities). We encourage contributions from the scientific community to further develop the generality and functionality of the framework.

5) Although the issue of novelty of the approach was raised by one of the reviewers, and this view is shared, it will likely not be helpful to try to rewrite the manuscript to stress the novelty. Taking steps to ensure the usefulness and then highlighting them will probably be more fruitful in terms of improving the manuscript.

Network-based parallel processing in and of itself is, of course, not novel. But the application of such a framework to provide experimental flexibility and high temporal precision while minimizing coding demands is unique. The unmet need for a system that jointly meets these competing goals is evident since there have been multiple previous attempts to develop such a system. However, existing alternatives have had limited success compared to the REC-GUI framework. We have further clarified this point and highlighted the usefulness of the framework in the Discussion.